# Examining the Influence of Authentic Leadership on Follower Hope and Organizational Citizenship Behavior: The Mediating Role of Follower Identification

**DOI:** 10.3390/bs13070572

**Published:** 2023-07-10

**Authors:** Kiho Jun, Zhehua Hu, Joonghak Lee

**Affiliations:** 1Faculty of Business and Management, BNU-HKBU United International College, Zhuhai 519087, China; kihojun@uic.edu.cn (K.J.); zhehuahu@uic.edu.cn (Z.H.); 2College of Business, Gachon University, Seongnam-Si 13120, Republic of Korea

**Keywords:** leadership, authentic leadership, identification with the supervisor, social identification, hope, organizational citizenship behavior

## Abstract

Authentic leadership’s influence on follower outcomes through the mediating roles of identification with the supervisor and social identification forms the core purpose of this research. By studying this less explored relationship within leadership studies, we aim to elucidate how these factors interrelate within the context of follower hope and organizational citizenship behavior (OCB). Using a quantitative methodology, we gathered and analyzed data from a sample of 241 employees across various South Korean businesses. Our main findings reveal that a follower’s identification with the supervisor significantly mediates the relationship between perceived authentic leadership and follower outcomes, such as hope and OCB. Concurrently, the study found that strengthening employee identification with their work group positively enhances these outcomes. From these findings, we conclude that authentic leadership can effectively drive follower identification, fostering beneficial outcomes, such as hope and OCB. It also suggests that workplaces that promote authentic leadership and a culture of strong supervisor and social identification can effectively enhance overall employee well-being and organizational performance.

## 1. Introduction

Authentic leadership has emerged as a robust strategy to combat trust deficits in businesses and individuals [1]. This approach has piqued the curiosity of organizational scholars and practitioners alike [2], resulting in two focal avenues in contemporary research. The first highlights the range of effects authentic leadership has on followers, spanning diverse outcomes, such as job satisfaction, organizational citizenship behavior (OCB), job performance, and work engagement [3,4,5,6,7]. The second stream is devoted to understanding the fundamental processes by which authentic leadership impacts these outcomes at both individual and organizational levels [8]. Notably, this exploration has elucidated the complex mechanisms linking authentic leadership with follower outcomes, incorporating factors, such as perceptions, cognition, attitudes, and behaviors [7,9,10,11,12].

However, despite considerable progress, there remain knowledge gaps in understanding how authentic leadership influences follower outcomes [8]. Recognizing this, we focus our investigation on the processes connecting authentic leadership with follower outcomes, underscoring the role of follower identification. Various studies have indicated that leadership behaviors can profoundly influence followers by shaping their identities [13,14,15]. Similarly, authentic leaders can stimulate their followers to align and connect not only with the leader but also with their organization [16]. Despite these findings, relatively few studies have investigated the mediating role of identification in the context of leadership [17,18].

To address this gap, our study explores identification-based mediators within the authentic leadership–follower outcomes framework. We place emphasis on follower identification with the supervisor and work group, considering their potential as key mediating factors [19]. Prior research suggests that authentic leadership can directly shape follower attitudes and behaviors [20]. Drawing from this insight, we propose that authentic leaders can influence follower identities, which then impact follower outcomes. In addition, our study explores the simultaneous role of identification with the leader and identification with the work group (i.e., social identification) in authentic leadership research [21,22]. To achieve a more comprehensive understanding of how authentic leaders influence follower outcomes, we introduce two intermediate variables: identification with the supervisor and social identification. We assert that these identification forms provide a more holistic explanation for the effects of authentic leadership on followers’ motivations and behaviors.

The subject of our study is the impact of perceived authentic leadership on follower outcomes, evaluated in a sample of 241 employees across different South Korean organizations. Guided by theories of relational identification and authentic leadership, we posit that identification with the supervisor serves as an intermediary mechanism linking authentic leadership to follower outcomes [13,23]. Furthermore, we suggest that authentic leadership can also impact follower outcomes via social identification, drawing upon social identity theory [24].

Focusing on follower hope and OCB as potential outcomes of authentic leadership, our research underscores the relevance of instilling hope within organizational settings [25,26]. Further, we underscore the potential of authentic leadership in fostering behaviors beyond job duties [27]. In conclusion, this study contributes to the existing literature in three ways: by introducing two types of identification as mediators in the authentic leadership–follower outcome relationship, by examining how authentic leadership enhances follower hope, and by testing the applicability of the association between authentic leadership and follower outcomes across different cultural contexts. Ultimately, our research aims to shed light on the complex, multifaceted interplay between authentic leadership and follower outcomes, thereby guiding future scholarly discourse and practical application. The research model was proposed as Figure 1.

## 2. Literature Review and Hypothesis Development

### 2.1. Authentic Leadership and Identification with the Supervisor

The concept of personal identification refers to an individual’s subjective experience of unity with another person [28]. In personal identification, “an individual identifies with the attributes of a target that make that target who he or she is—namely, his or her personal identities” [28] (p. 31). Within the context of leadership, personal identification (i.e., identification with the supervisor) is defined as the process whereby follower beliefs about a leader become self-defining and self-referential [29].

Various theories of leadership emphasize the role of such identification with the supervisor and argue that the influence of leaders depends on the extent to which their followers identify with them (e.g., [30,31]). For example, the charismatic leadership theory of Conger and Kanungo [32] emphasizes the vital role of personal identification in determining the influence that charismatic leaders have on their followers. Identification with the supervisor is also an important outcome of transformational leadership, which has been shown to strengthen follower personal identification by influencing follower self-concepts [29,33,34]. Lastly, authentic leadership theory also suggests that identification with the supervisor (i.e., personal identification) is a powerful mechanism linking authentic leadership to followers’ attitudes and behaviors (e.g., [1,2,20,35]).

Scholars suggest two theoretical frameworks to explain the relationship between authentic leadership and follower identification with the supervisor: (1) The fundamental principles of the theory of authentic leadership relate to the connection between authentic leadership and personal identification [20], and (2) authentic leadership behaviors possess the capacity to establish superior quality relationships with employees, thereby promoting greater identification of followers with the organization (i.e., relational identification theory) [36].

First, regarding the idea that authentic leadership serves as the foundation for various leadership styles, including but not limited to ethical and transformational leadership [37], scholars theorize how authentic leadership influences follower identification with the supervisor. Avolio et al. [20] argue that “since there is overlap between transformational and authentic leadership, authentic leaders are likely to stimulate follower’s personal identification (p. 806).” They further argue that in leading by example, such as by displaying high moral standards, honesty, and integrity, authentic leaders cause followers to identify with them. Leadership behaviors of this nature facilitate a strong bond between followers and their leaders, resulting in a gradual alignment of their respective values, beliefs, goals, and actions. Thus, followers identify with authentic leaders because “their core values are accessible so that followers can form self-defining relationships with them” [38]. Research shows that authentic leadership is significantly related to follower personal identification (see Lemoine et al. [2] for a review).

Second, scholars used insights from relational identification theory [36]. To understand the relationship between authentic leadership and follower identification. Sluss and Ashforth [36] introduced the notion of identifying with others at an interpersonal level pertains to the relationships individuals have with others within the context of their roles, such as the relationship between a supervisor and their subordinates. This theory refers to relational identification as the extent to which an individual defines himself or herself in terms of the leader–follower role relationship [36]. Scholars argue that relational identification depends on the attractiveness of the role relationship between the leader and the follower [39]. Our proposition suggests that the leadership style known as authentic leadership can bolster the sense of relational identification followers have with their leader, by elevating the perceived appeal of the relationship between them. The authentic leader achieves this through embodying characteristics of honesty, integrity, compassion, and high moral standards, thereby engendering trust and fostering superior quality relationships with their followers [20], leading to higher relational identification.

Drawing upon the two perspectives discussed above, first, we expect authentic leaders to make followers identify with the leader by recognizing that they share similar values with the leader (i.e., personal identification). Second, as a result of their perceived positive disposition and propensity to encourage transparency and responsibility in their interactions, authentic leaders exhibit behaviors that are linked to a greater degree of relational identification. Taking both arguments, we propose the following hypothesis:

**H1.** *The perception of authentic leadership behavior is positively related to follower identification with the supervisor*.

### 2.2. Mediating Effects of Identification with the Supervisor

Follower identification with the supervisor has been recognized as a “conduit through which leadership has many of its effects” [28] (p. 28). The impact of positive leadership styles, such as moral leadership and transformational leadership, is closely tied to followers’ personal identification with their leaders [40,41]. Scholars also have suggested that follower identification with the supervisor is likely to influence follower attitudes. For example, according to Qu et al. [39], the association between transformational leadership and follower creativity is mediated by the degree of personal identification that the follower feels toward the leader. Additionally, the relationship between follower personal identification and creativity is contingent upon the leader’s expectations regarding the level of creativity that the follower can achieve. Lian et al. [42] revealed that the connection between change-oriented leadership and the well-being of employees was influenced by the degree of identification that followers had with their leader. In a similar vein, Hobman et al. [43] demonstrated that the connection between supportive leadership and job satisfaction was moderated by the extent of personal identification followers had with their leader.

Scholars in authentic leadership have also suggested that authentic leaders may affect their followers to a greater degree when their followers identify with them more intensely [20]. Many have examined the intermediate psychological mechanisms linking authentic leadership to follower attitudes and behaviors. For instance, Wong et al. [21] demonstrated that personal identification is a key mechanism by which authentic leaders instill trust in their followers in their leadership. In addition, Lux et al. [38] found that authentic leadership influences follower personal identification, which can lead to positive follower outcomes (i.e., work engagement and job satisfaction) when mediated by their affective organizational commitment. Liu et al. [35] examined how authentic leadership encourages followers’ whistleblowing, focusing on the mediating role of personal identification. In their study, they found that personal identification played a significant mediating role in the relationship between authentic leadership and employee whistleblowing. A recent study dissected three intermediary mechanisms that elucidate how authentic leadership influences employees’ behavior in the workplace. This study found that supervisor identification is a direct repercussion of authentic leadership, as well as a mediating factor that facilitates the connection between authentic leadership and various aspects of employee behavior [44]. Notably, previous studies have highlighted the distinction between personal and organizational identification. For instance, personal identification has been investigated as the mechanism through which authentic leadership influences organizational identification, which subsequently leads to outcomes [45]. Additionally, the mediating role of personal identification was found to be less significant than organizational identification in military settings, where identification with leaders is traditionally viewed as crucial for success [46].

First, in line with these empirical findings, this study seeks to uncover the mechanisms by which authentic leadership affects follower hope. Authentic leadership theory and empirical studies suggest that followers can develop hope through their continuous interactions with authentic leaders ( e.g., [47,48,49,50]). This raises the following question: What underlying mechanisms determine the effects of authentic leaders on follower hope? Avolio et al. [20] suggested that follower hope can be more strongly developed when followers identify with hopeful and positive leaders, such as authentic leaders, because hopeful authentic leaders are more likely to motivate and actively engage followers by strengthening their personal identification. For this to occur, the goals of authentic leaders must be related to their followers’ self-structure [51]. In this study, we propose that identification with a more hopeful or positive leader increases follower hope. Specifically, in forming an attachment with such a leader, a follower is more likely to experience comfort rather than crisis and develop an intense feeling of stability and an ambitious orientation toward their goals. Consequently, this results in high follower motivation to accomplish their goals. Because hope is a highly positive motivational state, it is argued that strong follower identification with the leader has a positive association with follower hope.

Second, this current study suggests that the connection between authentic leadership and follower OCB is mediated by the extent to which the followers identify with their supervisor. Scholars have proposed that specific leadership styles, such as transformational leaders, influence their followers to exhibit higher levels of contextual performance (e.g., extra-role behavior) by serving as role models with whom followers are likely to identify themselves with the supervisor [52,53]. There have been assertions made by certain scholars that the association between transformational and charismatic leadership and contextual performance of followers can be attributed to their identification with the leader (i.e., extra-role performance, organizational citizenship behavior [31,54,55]. Many empirical studies have confirmed the mediating effects of follower identification with the leader on the relationship between leadership styles and different outcomes: dependency [34] and leader-directed OCB [14]. Specifically, Liu et al. [14] found that personal identification predicted employees’ extra-role behavior. Authentic leadership scholars also suggest that the impact of authentic leadership on followers might be very powerful through the identification with the supervisor (e.g., [20]). In other words, authentic leaders’ behavior can motivate followers to engage in more extra-role behaviors (i.e., OCB) by specific mechanisms, such as follower identification with the supervisor.

In accordance with the aforementioned theoretical and empirical findings, we propose that authentic leadership enhances follower OCB and that this relationship is mediated by follower identification with the supervisor. Extending the findings of prior research that examined the mediating role of personal identification in the relationship between authentic leadership and follower outcomes (e.g., [35]), we hypothesize that follower identification with the supervisor also positively influences the level of OCB because followers perceive the leader as an agent of the organization and a representative of their team [56]. Specifically, employees’ immediate supervisors in their team or work group are significant others in their daily lives in an organization. Thus, we argue that leaders’ behaviors may even influence employees’ behavior toward their work groups, organizations, and their immediate supervisors, through an increased level of identification with the leader. Recent reviews show that identification with an authentic leader is positively associated with follower motivation to engage in citizenship behavior [1,2]. Accordingly, we propose the following hypotheses.

**H2.** *The positive relationship between the perception of authentic leadership behavior and hope is mediated by follower identification with the supervisor*.

**H3.** *The positive relationship between the perception of authentic leadership behavior and organizational citizenship behavior is mediated by follower identification with the supervisor*.

### 2.3. Authentic Leadership and Social Identification

Drawing upon social identity theory, Tajfel and Turner [57] assert that behavior can be categorized as interpersonal behavior (behavior associated with acting as an individual) or intergroup behavior (behavior resulting from group membership). They argue that individual behavior is influenced by the level of social identification. For example, there exists a positive correlation between a stronger sense of social identification and the manifestation of behaviors that originate from an individual’s group membership [58].

Social identification refers to a person’s sense of belonging to a particular group. When individuals identify with a group, they partly base their self-concept on their belonging to that group [59]. Abundant research has shown that social identification affects follower behaviors that are typically associated with leadership effectiveness (for a review of this topic, see [60,61]). It has been suggested that the crux of charismatic and transformational leadership lies in the transformation of followers’ motivation from a self-centered focus to a collective orientation. For example, leader influence on follower social identification is central to the motivational theory of charismatic leadership proposed by Shamir et al. [62]. These authors suggested that charismatic and transformational leaders successfully influence their followers by connecting these followers’ self-concept to their organization’s mission and to that of the group. This causes followers to become self-expressive. Multiple studies have furnished proof of the correlation between different components of charismatic and transformational leadership and social identification. For instance, Shamir et al. [63] found that supportive leader behavior and emphasis on the collective identity were positively related to follower identification with their groups. By exhibiting idealized behaviors, transformational leaders can enhance the social identity of their followers; consequently, these followers are more likely to identify with their group [64].

Regarding the relationship between authentic leadership and social identification, many scholars also provide a strong support for the positive relationship between authentic leadership and social identification [19,20]. For example, Luthans and Avolio [65] proposed that authentic leaders’ core task is to find followers’ strengths and help build them appropriately and then link them to a common mission by enhancing identification. Scholars also suggested that authentic leaders can enhance their followers’ engagement by strengthening group members’ identification with their work groups and with their organizations. Specifically, authentic leaders “increase their followers’ social identification by creating a deeper sense of high moral values and expressing high levels of honesty and integrity in their dealings with followers” [20] (p. 807). Drawing from discussions from various leadership theories, thus, we propose that authentic leaders significantly affect their followers’ social identification by displaying high levels of honesty and integrity in their relationships with their followers. That is, followers relate to the values and moral standards of their work groups and finally strongly identify themselves with their groups.

**H4.** *The perception of authentic leadership behavior is positively related to follower social identification*.

### 2.4. Mediating Effects of Social Identification

In this study, social identification constitutes an additional psychological intermediary (apart from personal identification) through which authentic leadership affects follower hope and extra-role behaviors. Scholars have suggested that the effects of leader behaviors, such as those of authentic leaders, on their followers’ psychological states (i.e., hope) and behaviors (i.e., OCB) can be facilitated by specific intermediate processes, such as follower social identification with the respective work groups [1,18]. van Knippenberg et al. [15] (p. 831) also suggested that “the available evidence supports the proposition that leadership may affect follower identification with the collective, and this effect on identification mediates the effects of leadership on follower attitudes and behavior.” Prior research has shown that leader behaviors affect employees via follower identification with the team or work group. For example, Cicero and Pierro [66] found that follower identification with the work group (i.e., social identification) mediated the relationship between charismatic leadership and employee well-being. Moreover, it has been proposed that identification with an authentic leader and associated group displaying high levels of transparency, integrity, and moral standards elevates the levels of positive follower psychological states, such as their hope and optimism (e.g., [67]). In addition to positive organizational outcomes, a recent study shows that social identification could explain the mechanism by which authentic leadership influences unethical pro-organizational behaviors [68].

Social identity theory posits that when individuals identify strongly with a group, they become deeply concerned about the group’s welfare. Consequently, their behaviors become oriented toward benefitting the interests and needs of the group [24]. Studies have provided evidence that the behavior of individuals within a group is influenced by their social identity. Blader and Tyler’s [24] research serves as an example, as it revealed a correlation between employees’ social identity and their supervisor-rated OCB. Therefore, social identification may explain how authentic leaders reinforce group identification, thereby promoting hope and OCB among their followers. Empirical studies have shown that authentic leaders may evoke social identification among their followers, which in turn mediates the effects of these leaders’ behaviors on their followers (e.g., enhancing their followers’ OCB and other attitudes). For example, Edú-Valsania et al. [22] showed that follower identification with the workgroup mediated the effect of authentic leadership on their knowledge-sharing behaviors. Wong et al. [21] found that by enhancing follower social identification, authentic leaders positively affected the psychological states and behaviors of their followers.

The processes that shape social identification [69] elicit a sense of oneness with the group; this induces individuals to perceive the group’s goals, interests, and norms as their own [70]. In other words, identification with the workgroup is expected to motivate team-oriented behavior (e.g., OCBI). Supporting this notion, empirical studies have shown that a strong collective identity orients team members toward the pursuit of a collective (rather than an individual) goal [71] and generates more cooperative work behaviors [72]. Individuals who exhibit a strong sense of identification with their team are likely to exhibit greater motivation to engage in behaviors that benefit the team as a whole, as a result of their convictions regarding the significance of pursuing objectives that serve the collective interests of the group, rather than solely serving their own individual goals [71]. In this study, we also argue that followers also tend to engage in leader-directed OCB if they strongly identify themselves with their work groups because their supervisor is considered a representative of their groups. Thus, social identification is expected to motivate group members to invest in a collective goal, such as helping other group members and their leaders. Hence, we hypothesize that authentic leadership enhances social identification, which in turn increases follower hope and OCB. Specifically, we propose the following hypotheses:

**H5.** *The positive relationship between the perception of authentic leadership behavior and hope is mediated by follower social identification*.

**H6.** *The positive relationship between the perception of authentic leadership behavior and organizational citizenship behavior is mediated by follower social identification*.

## 3. Methodology

### 3.1. Data Collection and Sample

Our research population comprised 241 full-time employees hailing from a multitude of businesses throughout Korea. The use of a convenience sampling strategy was applied, due to its practicality and facilitation of potential participant access. The chosen population incorporated 300 professionals employed in various South Korean establishments, including financial institutions, airlines, and research centers, among others. The voluntary nature of participation was stressed to the participants, along with the assurance that non-participation would not yield negative repercussions, as indicated by Podsakoff et al. [73]. Moreover, participants were assured of the confidentiality of their responses, with the data collected exclusively reserved for research purposes. Each participant’s consent was acquired before conducting the survey, which was administered using traditional pen-and-paper methods.

The exploration of the influence of perceived authentic leadership of supervisors on subordinate outcomes was the main objective of our study, employing a self-report research methodology. Participants were requested to deliver self-reports concerning their perception of several critical variables, namely authentic leadership, level of identification, hope, and organizational citizenship behavior. To curb the potential problem of common method variance inherent in self-report research paradigms, we utilized a time-lagged research model, which entailed a two-phase survey, conducted at two separate intervals three weeks apart. This methodology aligns with the proposition of Donaldson and Grant-Vallone [74]. The initial phase involved disseminating 300 hard-copy questionnaires, soliciting participants’ perception of authentic leadership, level of identification, and other demographic specifics (Time 1). An explanatory note accompanied the questionnaire at Time 1, elucidating the research intent, confidentiality, anonymity, and the voluntary nature of the study. A total of 280 questionnaires were retrieved (accounting for a 93.33% response rate). During the subsequent phase, 280 physical questionnaires were circulated among those who participated in Phase 1. Here, participants evaluated their hope and organizational citizenship behaviors over the prior weeks (Time 2). The returned questionnaires amounted to 250, corresponding to an 89% response rate. After eliminating nine incomplete surveys, 241 fully completed surveys remained for further analysis, translating to an effective response rate of 80.3%.

Out of the final 241 participants, a dominant 67.7% were identified as males. Participants were also classified based on their organizational rank, inclusive of standard employees (36.9%), deputy section chiefs (25.8%), section leaders (19.5%), and either department heads or board members (17.8%). The participant tenure varied, with a plurality of over 7 years (35.4%), followed by more than 5 years but less than 7 years (17.1%), more than 3 years but less than 5 years (23.3%), and less than 3 years (24.2%) in their respective organizations. Furthermore, participant affiliations comprised of organizations with over 2000 employees (73.7%), between 1000 and 2000 employees (4.7%), between 500 and 1000 employees (2.5%), and less than 500 employees (19.1%).

### 3.2. Measurements

We translated the original English survey items into Korean to cater to the participants. The translation–back-translation method [75] was used to confirm the translation’s accuracy. To validate the lucidity and reliability of the questionnaire items, a preliminary trial was conducted with 20 participants from one of the organizations taking part in the study before the final distribution of the survey. The pilot test did not disclose any significant comprehension issues concerning the items in the questionnaire. Except where stated otherwise, all questions required the respondents to indicate their responses on a five-point Likert scale (1 = strongly disagree, 5 = strongly agree). Detailed measurement scales were listed in Appendix A.

*Authentic leadership.* To gauge the perception of genuine leadership, sixteen items formulated by Walumbwa et al. [76] were employed. These questions sought to capture the respondents’ perception of their immediate supervisor’s authenticity in their leadership. Respondents were asked to assess their direct manager’s authentic leadership behavior via a set of sixteen parameters. An exemplar item is my supervisor communicates precisely what he/she intends (α = 0.94).

*Identification with Supervisor and Social Identification.* Six items derived from the metrics of organizational identification developed by Mael and Ashforth [59] and Shamir et al. [63] were adjusted to quantify follower identification with the supervisor. An example item is when someone applauds my superior, I feel personally complimented (α = 0.91). Social identification was evaluated employing six items similar to those used to gauge personal identification. An exemplar item includes this department’s triumphs are my triumphs (α = 0.89).

*Hope.* Follower hope was quantified using six items crafted by Snyder et al. [77]. An exemplar item is if I should find myself in a predicament, I could conceive of numerous solutions to extricate myself from it (α = 0.82).

*Organizational Citizenship Behavior.* In this study, we focus on two targets of follower OCB to test different mediating mechanisms: OCB targeted at work groups and OCB targeted at leaders. Specifically, the first type of OCB benefits specific individuals, such as work group members, whereas the second OCB directly benefits the supervisor. Therefore, OCB was evaluated using four items from Wayne et al. [78]. An exemplar item is I initiate the orientation of new employees to the department, even if it falls outside of my official job responsibilities (α = 0.73).

*Control variables.* Potential factors that might offer alternative explanations for the hypothesized relationships among the variables were considered [26,79]. These variables included sex, rank, organizational tenure, firm size (measured by employee count), and job stress. In this study, the sex variable was numerically coded with “1” for “male” and “2” for “female”. Similarly, numerical values were used for rank with “1” representing “ordinary employees”, “2” for “deputy section chiefs”, “3” for “section heads”, and “4” for “department heads or board members”. Organizational tenure, firm size, and job stress were encoded similarly. Job stress was evaluated with a four-item scale developed by Keller [80].

## 4. Results

The study variables underwent a series of confirmatory factor analyses (CFA) to scrutinize their factor structure and to aid the subsequent correlation and regression analyses. The outcomes of the CFA are displayed in Table 1. Initially, a CFA for the five constructs (i.e., authentic leadership, identification with the supervisor, social identification, hope, and OCB) was performed using AMOS 21. Prior to hypothesis testing, the congruence of the proposed five-factor model was evaluated. Based on the results showcased in Table 1, the fit indices demonstrated a good model fit (χ^2^/df = 1.66, RMSEA = 0.056, NFI = 0.82, IFI = 0.92, TLI = 0.91, and CFI = 0.92). According to the criteria established by Hair et al. (2010), factor loadings less than 0.5 were discarded (AL4; AL5). Subsequently, a four-factor model was appraised, wherein social identification and identification with the supervisor were amalgamated (χ^2^/df = 2.65, RMSEA = 0.09, NFI = 0.70, IFI = 0.79, TLI = 0.78 and CFI = 0.79). Thirdly, a three-factor model was tested by combining social identification, identification with the supervisor, hope, and OCB (χ^2^/df = 2.91, RMSEA = 0.10, NFI = 0.67, IFI = 0.76, TLI = 0.74 and CFI = 0.76). Ultimately, a binary component structure was investigated, in which all survey items, except for those relating to authentic leadership, were loaded onto a singular factor (χ^2^/df = 3.37, RMSEA = 0.11, NFI = 0.62, IFI = 0.7, TLI = 0.68 and CFI = 0.7). Overall, the fit indices suggested that the hypothesized five-factor model fits better than the other models.

Table 2 delineates the detailed descriptive statistics and interconnections of the variables, illustrating that there existed a positive association between authentic leadership and hope (r = 0.39, *p* < 0.01), as well as OCB (r = 0.21, *p* < 0.01). Personal identification was significantly positively associated with authentic leadership (r = 0.65, *p* < 0.01), hope (r = 0.51, *p* < 0.01), and OCB (r = 0.33, *p* < 0.01). Lastly, social identification demonstrated notable positive ties with authentic leadership (r = 0.37, *p* < 0.01) and hope (r = 0.50, *p* < 0.01), also showing a positive association with OCB (r = 0.34, *p* < 0.01).

For analyzing the main effects of authentic leadership (H1 and H4), we employed the OLS regression analysis. Initially, the analysis manifested a robust positive association between supervisor identification and authentic leadership (β = 0.62, *p* < 0.001, SE = 0.06), which supported H1. Furthermore, it exhibited a positive association between authentic leadership and social identification (β = 0.37, *p* < 0.001, SE = 0.06), supporting H4.

To explore the mediation effects, we adhered to the methodology propounded by Kenny et al. [81]. Initially, the analysis illustrated that the impact of authentic leadership on hope vanished when supervisor identification was incorporated in Model 2 of Table 3, compared to Model 1 (β = 0.11, n.s.). Consequently, supervisor identification entirely mediated the relationship between authentic leadership and hope, backing H2. Secondly, the analysis exhibited that when supervisor identification was included in Model 3 of Table 3, the influence of authentic leadership on OCB vanished as compared to Model 3 (β = −0.02, n.s.). As such, supervisor identification wholly mediated the relationship between authentic leadership and OCB, thereby supporting H3. Thirdly, when social identification was incorporated into Model 2 of Table 4, the influence of authentic leadership dwindled compared to Model 2 (β = 0.37, *p* < 0.001, β = 0.23, *p* < 0.01).

This outcome signifies that social identification partially mediated the relationship between authentic leadership and hope, hence H5 was corroborated. Fourthly, the analysis found that the impact of authentic leadership on OCB dissipated in Model 3 of Table 4 when social identification was considered (β = 0.08, n.s.). Thus, supervisor identification completely mediated the relationship between authentic leadership and OCB, endorsing H6.

To authenticate the mediation effect, we carried out a mediation test based on bootstrapping. Table 5 presents the findings of the mediation analysis, which constituted a bootstrap-based mediation test executed utilizing the PROCESS macro [82] in SPSS 25. This specific method’s usage is heavily favored over other procedures, such as those specified by Baron and Kenny [83], due to its superior statistical efficacy, as evidenced by prior research [84]. Here, the importance of the estimated indirect effect was evaluated using 5000 bootstrap samples and a bias-corrected 95% confidence interval (CI). The indirect effect was deemed significant when the bias-corrected 95% CI excluded zero [85].

The bootstrap analysis results revealed that the indirect effect of authentic leadership on hope via supervisor identification was significant (indirect effect = 0.18, SE = 0.04, 95% CI = 0.11 to 0.27; direct effect = 0.05, SE = 0.06, 95% CI = −0.07 to 0.18; total effect = 0.24, SE = 0.05, 95% CI = 0.13 to 0.34), endorsing H2. Moreover, as presented in Table 5, the indirect effect of authentic leadership on OCB via supervisor identification was significant (indirect effect = 0.15, SE = 0.05, 95% CI = 0.06 to 0.24; direct effect = −0.06, SE = 0.07, 95% CI = −0.20 to 0.08; total effect = 0.09, SE = 0.06, 95% CI = −0.03 to 0.21), thereby backing H3. The same mediation test employed to evaluate H2 and H3 was also utilized for H5 and H6. The bootstrap analysis outcomes exhibited that authentic leadership influenced employee hope through social identification, thereby corroborating H5 (indirect effect = 0.11, SE = 0.03, 95% CI = 0.05 to 0.19; direct effect = 0.13, SE = 0.05, 95% CI = 0.03 to 0.23; total effect = 0.24, SE = 0.05, 95% CI = 0.14 to 0.34). The outcomes also displayed that authentic leadership influenced employee OCB through social identification (indirect effect = 0.09, SE = 0.03, 95% CI = 0.04 to 0.17; direct effect = -.005, SE = 0.06, 95% CI = −0.12 to 0.11; total effect = 0.09, SE = 0.06, 95% CI = -.03 to 0.21), hence backing H6.

## 5. Discussion

This study investigated the relationship between authentic leadership perception and follower outcomes, such as hope and OCB, by focusing on the mediating roles of follower identification with the supervisor and social identification. The results showed that follower identification with the supervisor and social identification are important mediators in the relationship between authentic leadership perception and follower outcomes.

### 5.1. Theoretical Contributions

The findings extend the existing literature in four ways. First, we draw on the identification literature to explain how leaders’ perceived authenticity influences their followers’ psychological states and behaviors. Although identification has been shown to be a key factor influencing this process, owing to its effects on individuals’ motivations and behaviors (e.g., personal identification [28], social identification [15], and organizational identification [86], it remains understudied in the literature on authentic leadership [1]. In particular, few studies have simultaneously included identification with the supervisor and social identification in their empirical or theoretical models. By contrast, the findings of our investigation indicate that authentic leaders possess the ability to establish robust and positive exchange relationships with their followers, which, in turn, contributes to the enhancement of both follower identification with their supervisors and social identification.

In addition to previous research on the underlying mechanisms linking authentic leadership and its outcomes [1,20], our study reveals that identification with the supervisor and social identification play a significant mediating role in elucidating how the perception of authentic leadership influences both follower hope and organizational citizenship behavior (OCB). Prior studies have examined various mechanisms through which authentic leadership perception impacts follower outcomes. For instance, the presence of trust in the leader has been identified as a positive factor in the relationship between authentic leadership and follower outcomes [4,10]. Scholars have also discovered that authentic leadership indirectly influences team performance [87]. As with previous findings on other leadership constructs, such as transformational leadership and authentic leadership [18,34], the findings of our study suggest that follower identification with the supervisor may be a critical intervening variable that links leadership to follower outcomes. In addition, our study contributes to the research on social identity and authentic leadership by providing additional evidence demonstrating the mediating role of follower social identification in the relationship between leadership behaviors and follower outcomes [88,89]. Specifically, we integrated social identity theory and authentic leadership theory to examine the effects of authentic leadership perception on follower outcomes (i.e., hope and OCB) by showing that these two types of identification differently mediate the effects of leadership on followers.

Third, the results of our study reveal that authentic leaders can develop follower hope by enhancing follower identification with the supervisor and social identification [20]. To date, no studies have investigated how authentic leadership can increase the level of follower hope. Therefore, our findings answer this important question about follower hope, which is a key motivational construct that can be used to create roadmaps to guide an individual toward a goal. Given the importance of hope at work, our study contributes by improving the understanding of how specific types of leader behaviors enhance follower hope.

Lastly, our study, which utilized data collected from multiple organizations in Korea, has established a noteworthy association between the perception of authentic leadership and follower outcomes. Previous research has shown that authentic leadership can have a positive impact on employees and, consequently, lead to favorable follower outcomes [2]. Nonetheless, scholars have emphasized the need for further research to examine the generalizability of such findings in diverse contexts [2,7,90]. Our study has contributed to the existing literature on authentic leadership by confirming the previous research results using data from diverse workforces in various organizations within a single Asian country. Therefore, our study has contributed to the general understanding of the relationship between authentic leadership and follower outcomes.

### 5.2. Practical Implications

The practical findings of this study underscore the potency of authentic leadership in fostering follower identification and, consequently, desirable organizational outcomes. It highlights the need for innovative leadership development tools that primarily focus on nurturing authenticity. By enabling leaders to establish stronger relational ties, these tools can foster positive follower behaviors, enhancing overall productivity and workplace harmony.

Moreover, the findings emphasize the importance of open dialogues between managers and employees, as such interactions often lead to higher levels of identification, a precursor to organizational success. Authentic leaders who actively engage in conversations with their teams not only strengthen mutual bonds but also stimulate a sense of belonging and commitment among the workforces.

Our research elucidates the role of hope as a crucial element in the workplace, corroborating prior findings [26] on its positive influence on employee performance. Authentic leaders can nurture hope and resilience among employees, particularly relevant in the context of post-pandemic recovery. In an era marred by unusual workplace circumstances [91,92], our findings suggest that authentic leadership can enhance employee motivation, thereby improving organizational performance.

Further, the study underscores the mediating roles of identification with the supervisor and social identification between authentic leadership and follower outcomes. By highlighting these mediating influences, we gain a more profound understanding of how authentic leadership shapes follower outcomes. The study’s findings can potentially revolutionize the organizational psychology literature by offering insights into the dynamics of leadership and its impact on followers.

Lastly, while our study focuses on South Korean business organizations, the insights drawn have broader geographical implications. The lessons learned about authentic leadership and its role in enhancing follower identification can apply to countries with diverse cultural norms, such as Japan, China, or Singapore. The emphasis on relational and social identification in authentic leadership spotlights the necessity for leaders to adapt their strategies to the cultural context, particularly in multinational operations. This adaptability could further foster positive employee outcomes across various cultural backgrounds.

### 5.3. Limitations

This research exhibits certain limitations. Initially, common method bias was inevitable as the study relied on followers’ self-reports to measure all variables. In order to mitigate this issue, the researchers took various measures to minimize the common method variance prior to data collection. For instance, a time delay was introduced between the measurement of independent and outcome variables. Anonymity and confidentiality of participants were also ensured. Finally, clear instructions were provided for completing surveys, along with definitions to prevent any confusion among participants [93]. Furthermore, the usage of self-reported data is recognized as a valid approach for evaluating perceptual outcomes and internal states, such as feelings and perceptions [94]. Given that several of the survey questions asked employees about their personal perceptions (e.g., hope), the use of self-reported data may not be a grave issue in this study. Regarding OCB measurement, Vandenberg et al. [95] suggested that the OCB measurement from bosses might be biased; thus, self-reporting of OCB might be preferable for organizational behavior research. Moreover, Ilies et al. [96] argued that self-reports of OCB are more useful than others’ reports in that self-reports can exactly measure some aspects of OCB (especially OCBI) that are not observed in others’ reports. Taken together, the use of self-report measures in this study might not be a severe problem. However, future studies may use more refined methods (e.g., data from diverse sources and methods) to overcome common method bias.

Second, social identification was assumed to be an individual-level construct in the present study because this collective phenomenon has primarily been studied at the individual level (e.g., [97]). However, given that most individuals are nested in teams, studies have also highlighted the relevance of investigating social identification at the group level in addition to that at the individual level (e.g., [98]). For example, one study measured work group identification through group-level perspectives in the relationship between transformational leadership and follower outcomes [43]. Therefore, additional studies are needed to investigate the mediating effect of social identification at the group level on the relationship between authentic leadership and follower outcomes.

Third, the cross-sectional research design of our study is another limitation. Although the results confirmed all of the study hypotheses, the causal results should be interpreted with caution. For example, it is conceivable that followers with high personal identification perceive their leaders more favorably, causing them to evaluate these leaders as more authentic. To draw more robust conclusions about the cause-and-effect relationship between authentic leadership and follower outcomes, future studies should employ a longitudinal research design. Such an approach can provide a more comprehensive understanding of the dynamic nature of this relationship over time.

## 6. Conclusions

Our study highlights the importance of follower identification with the supervisor and workgroup in determining the effects of authentic leaders on follower outcomes. Compared with other leaders, authentic leaders can be more successful role models for their followers, and thus, they may better develop follower hope and encourage followers to behave favorably toward their coworkers. Notwithstanding some limitations, our study contributes to the growing literature on authentic leadership by testing specific psychological factors (i.e., identification with the supervisor and social identification) that may explain the effects of authentic leaders on follower outcomes.

## Figures and Tables

**Figure 1 behavsci-13-00572-f001:**
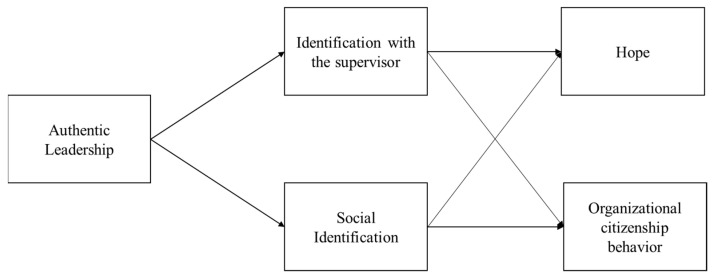
Research Model.

**Table 1 behavsci-13-00572-t001:** Measurement model and confirmatory factor analysis.

Factors	χ^2^	DF	χ^2^/DF	RMSEA	NFI	IFI	TLI	CFI
Five-factor model	938.86	565	1.66	0.06	0.82	0.92	0.91	0.92
Four-factor model	1562.15	589	2.652	0.09	0.70	0.79	0.78	0.79
Three-factor model	1723.38	592	2.911	0.10	0.67	0.76	0.74	0.76
Two-factor model	1998.68	594	3.365	0.11	0.62	0.70	0.68	0.70

**Table 2 behavsci-13-00572-t002:** Descriptive statistics and correlations.

	Mean	SD	1	2	3	4	5	6	7	8	9	10
1. Sex	1.32	0.47	1									
2. Rank	2.33	1.42	−0.31 **	1								
3. Tenure	2.64	1.20	−0.24 **	0.60 **	1							
4. Firm size	3.33	1.19	−0.05	−0.26 **	−0.1	1						
5. Job stress	3.44	0.68	−0.06	−0.04	−0.1	0.18 **	1					
6. Authentic leadership	3.28	0.77	−0.18 **	0.02	0.08	−0.12	−0.08	1				
7. Identification with the supervisor	2.87	0.85	−0.20 **	0.16 *	0.18 **	−0.09	−0.09	0.65 **	1			
8. Social identification	3.66	0.74	−0.15 *	0.31 **	0.24 **	−0.06	0.02	0.37 **	0.60 **	1		
9. Hope	3.45	0.58	−0.17 *	0.34 **	0.23 **	−0.23 **	−0.13 *	0.39 **	0.51 **	0.50 **	1	
10. OCB	3.55	0.60	−0.17 *	0.18 **	0.17 *	−0.07	−0.04	0.21 **	0.33 **	0.34 **	0.42 **	1

Note(s): * *p* < 0.05, ** *p* < 0.01.

**Table 3 behavsci-13-00572-t003:** Regression analyses for mediation test (Mediator= Identification with the Supervisor).

Variables	Model 1: DV = Identification with the Supervisor			Model 2: DV = Hope			Model 3: DV = OCB		
	B	SE B	β	B	SE B	β	B	SE B	β
Authentic leadership	0.69	0.06	0.62 ***	0.09	0.06	0.11	−0.01	0.07	−0.02
Identification with the supervisor				0.27	0.05	0.39 ***	0.22	0.06	0.31 ***
Adjusted *R^2^*		0.41			0.34			0.11	

Note(s): *** *p* < 0.001.

**Table 4 behavsci-13-00572-t004:** Regression analyses for mediation test (Mediator = Social Identification).

Variables	Model 1: DV = Social Identification			Model 2: DV = Hope			Model 3: DV = OCB		
	B	SE B	β	B	SE B	β	B	SE B	β
Authentic leadership	0.37	0.06	0.37 ***	0.18	0.05	0.23 **	0.06	0.06	0.08
Social Identification				0.27	0.05	0.35 ***	0.2	0.06	0.25 **
Adjusted *R^2^*		0.22			0.35			0.10	

Note(s): ** *p* < 0.01, *** *p* < 0.001.

**Table 5 behavsci-13-00572-t005:** Summary of direct and indirect effects.

Paths and Effects	Estimates	SE	95% Confidence Intervals
			LL~UL
**Direct Effects**			
Authentic Leadership → Identification with the supervisor	0.68	0.07	[0.550–0.820]
Authentic Leadership→ Social Identification	0.31	0.07	[0.180–0.450]
**Indirect Effects**			
Authentic Leadership → Identification with the supervisor → Hope	0.18	0.04	[0.110, 0.270]
Authentic Leadership → Identification with the supervisor → OCB	0.15	0.05	[0.062, 0.243]
Authentic Leadership → Social Identification → Hope	0.11	0.03	[0.055, 0.184]
Authentic Leadership → Social Identification → OCB	0.09	0.03	[0.039, 0.166]

Note(s): N = 241, LL = lower limit, UL = upper limit, S.E. = standard error.

## Data Availability

Data is unavailable due to privacy or ethical restrictions.

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
