# Peer review of "Examining the Influence of Authentic Leadership on Follower Hope and Organizational Citizenship Behavior: The Mediating Role of Follower Identification"

_behavsci, 2023, doi:10.3390/bs13070572_

Round 1

Reviewer 1 Report

The study presents a very interesting research question that explores the underlying process of authentic leadership’s influence on follower’s outcomes. I enjoyed reading the paper very much. In the meantime, I also would like to share my observations of the manuscript and hope would find them helpful.

1.       Conceptual Development: the authors have done a really good job laying out the overall structure of the paper (with the subtitles). What could be improved are:

a.       Theoretical development of how and why authentic leadership can impact follower’s identification and outcomes: You often reason the proposed relationships between authentic leadership and follower outcomes (including mediating processes) by citing previous studies on the relationship between either a general leadership (in Introduction) and its outcomes or other types of leadership and outcomes (such as charismatic or transformational leadership; page 5-7). This is not a convincing way to establish the conceptual links to help address your research questions. Authentic leadership is its own unique style and has its unique impact on followers. You should do a better job to articulate what this unique style of leadership is and how it behaves (different from charismatic and transformational leadership although with overlapping). You should also cite previous studies on specifically authentic leadership and following outcomes, instead of other types of leadership).

b.       Differentiating two identifications and their respective pathways: I do agree with the value in testing both the identification with leader (individual, “I” in OCB) and identification with the group/team or organization (social identification, “O” in OCB) simultaneously. However, I am not sure you have articulated the constructs clearly especially for the identification with group/team or organization.

Page7: identification with the team or group means identify with the team or group (e.g., department as you operationalized) as a whole, using the words like, my team, my organization etc. It is not the same as “identification with each other.” It is also not correct to say that it is how “followers collectively relate to the values and moral standards of their leaders with who they identify”, which would be in effect the consensus among followers in terms of their identification with the leader (“I”).

Page 8: if you wanted to differentiate between OCB-I and OCB-O, you should move the second paragraph to page 6, where you also talk about OCB. Since you only use one scale to measure all the OCBs for all  hypotheses, readers would need to know how you conceptualize them consistently. In addition, OCB-I would mean how followers’ OCB is targeted at the leader, OCB-O would mean how followers’ OCB is targeted at the team or department (should not be organization since you did not operationalize so). You writing should stay consistent with the empirical session (operationalization). I have more to share on this below.

2.       Measurement alignment with conceptual development:

In your conceptual section, you seem to argue that followers’ OCB would have different target, either the individual or team/organization. There should be an alignment (as the literature says) between the identification and OCB target:

For instance, identification with leader often leads to OCB targeted at the leader as well. Or, as you said, when followers see leader as the extension of the organization, they might extend their OCB to organization as well. Identification with group likely leads to group-targeted OCB. And your operationalization (scales) should reflect such alignment. But I am not sure this is the case. For instance, the OCBI, is a mixed of both O and I-targeted scale.

Proof read needed.

Author Response

We want to first thank you and the Reviewer for providing us with the opportunity to revise and resubmit our research. It is clear that you and the reviewers gave this manuscript (behavsci-2470103) very careful attention, and we have tried to reciprocate by addressing all of the various comments to the fullest extent possible. Below we have reproduced reviewers’ feedback in bold italics, followed by our response (blue color) in regular font. We carefully read the three reviewers’ comments and addressed and/or responded to each comment. We greatly appreciate this opportunity to revise and resubmit our manuscript. We welcome your continued feedback as we strive to make our manuscript as helpful to the field as possible.

Reviewer: 1

Recommendation:

The study presents a very interesting research question that explores the underlying process of authentic leadership’s influence on follower’s outcomes. I enjoyed reading the paper very much. In the meantime, I also would like to share my observations of the manuscript and hope would find them helpful.

Comments:

  1. Conceptual Development: the authors have done a really good job laying out the overall structure of the paper (with the subtitles). What could be improved are:

  • Theoretical development of how and why authentic leadership can impact follower’s identification and outcomes: You often reason the proposed relationships between authentic leadership and follower outcomes (including mediating processes) by citing previous studies on the relationship between either a general leadership (in Introduction) and its outcomes or other types of leadership and outcomes (such as charismatic or transformational leadership; page 5-7). This is not a convincing way to establish the conceptual links to help address your research questions. Authentic leadership is its own unique style and has its unique impact on followers. You should do a better job to articulate what this unique style of leadership is and how it behaves (different from charismatic and transformational leadership although with overlapping). You should also cite previous studies on specifically authentic leadership and following outcomes, instead of other types of leadership).

  • Differentiating two identifications and their respective pathways: I do agree with the value in testing both the identification with leader (individual, “I” in OCB) and identification with the group/team or organization (social identification, “O” in OCB) simultaneously. However, I am not sure you have articulated the constructs clearly especially for the identification with group/team or organization.

  • Page7: identification with the team or group means identify with the team or group (e.g., department as you operationalized) as a whole, using the words like, my team, my organization etc. It is not the same as “identification with each other.” It is also not correct to say that it is how “followers collectively relate to the values and moral standards of their leaders with who they identify”, which would be in effect the consensus among followers in terms of their identification with the leader (“I”).

  • Page 8: if you wanted to differentiate between OCB-I and OCB-O, you should move the second paragraph to page 6, where you also talk about OCB. Since you only use one scale to measure all the OCBs for all hypotheses, readers would need to know how you conceptualize them consistently. In addition, OCB-I would mean how followers’ OCB is targeted at the leader, OCB-O would mean how followers’ OCB is targeted at the team or department (should not be organization since you did not operationalize so). You writing should stay consistent with the empirical session (operationalization). I have more to share on this below.

Response to reviewer: We greatly appreciate your suggestion on how to develop more robust hypotheses about the relationship among key variables in this study. In the below section, we discuss how we did address your concerns about conceptual development.

First, about the theoretical development, we tried to discuss how authentic leadership affects identification, and then leads to follower outcomes using unique aspects of authentic leadership theory, instead of using other leadership theories. We added some key points to the manuscripts as follows. We also found some recent papers about the relationship between authentic leadership and follower outcomes.

“Lastly, authentic leadership theory also suggests that identification with the supervisor (i.e., personal identification) is a powerful mechanism linking authentic leadership to followers’ attitudes and behaviors (e.g., Avolio et al., 2004; Gardner et al., 2011; Liu et al., 2015; Lemoine et al., 2019). (Page 3)”

“Authentic leadership scholars also suggest that the impact of authentic leadership on followers might be very powerful through the identification with the supervisor (e.g., Avolio et al., 2004). In other words, authentic leaders’ behavior can motivate followers to engage in more extra-role behaviors (i.e., OCB) by specific mechanisms such as follower identification with the supervisor. (Page 5)”

Second, about your points for the clarity of social identification construct, we revised it clearly by focusing on work group and leaders who lead followers’ work group. We developed our hypothesis using authentic leadership theory, instead of other leadership theories. We revised the parts as follows.

“Regarding the relationship between authentic leadership and social identification, many scholars also provide a strong support for the positive relationship between authentic leadership and social identification (Avolio et al., 2004; Avolio et al., 2009). For example, Luthans and Avolio (2003) proposed that authentic leaders’ core task is to find followers’ strengths and help build them appropriately, and then link them to a com-mon mission by enhancing identification. (Page 6)”

Third, about your points on page 7, we deleted some unclear parts and revised it more clearly as follows. “Scholars also suggested that authentic leaders can enhance their followers’ engagement by strengthening group members’ identification with their work groups and with their organizations. (Page 6)”

“That is, followers relate to the values and moral standards of their work groups and finally strongly identify themselves with their groups. (Page 6)”

Fourth, about points you raised on differentiation of OCB, we radically changed our rationale and revised methodology part accordingly. We first deleted the part on page 6, and moved it to method part. Then, we added the explanations why we used four OCB items in the section of measures as follows.

“In this study, we focus on two targets of follower OCB to test different mediating mechanisms: OCB targeted at work groups and OCB targeted at leaders. Specifically, first type of OCB benefits specific individuals such as work group members, whereas the second OCB directly benefits the supervisor. Therefore, OCB was evaluated using four items from Wayne et al. (1997). (Page 8)”

In addition, to be consistent in terms of theoretical arguments and operationalization, we added some explanations about why we only focused on two targets of OCB (OCB toward the work group and OCB toward the leader) in the parts of mediating hypotheses as follows.

“Extending the findings of prior research that examined the mediating role of personal identification in the relationship between authentic leadership and follower outcomes (e.g., Liu et al., 2015), we hypothesize that follower identification with the supervisor also positively influences the level of OCB because followers perceive the leader as an agent of the organization and a representative of their team (Rhoades & Eisenberger, 2002). Specifically, employees’ immediate supervisors in their team or work group are significant others in their daily lives in an organization. Thus, we argue that leaders’ behaviors may even influence employees’ behavior toward their work groups, organizations, and their immediate supervisors, through an increased level of identification with the leader. (Page 5)” (for personal identification)

“In this study, we also argue that followers also tend to engage in leader-directed OCB if they strongly identify themselves with their work groups because their supervisor is considered a representative of their groups. Thus, social identification is expected to motivate group members to invest in a collective goal such as helping other group members and their leaders. (Page 7)” (for social identification)

Finally, we changed “OCBI” into “OCB” throughout the manuscript to be consistent.

  1. Measurement alignment with conceptual development:

In your conceptual section, you seem to argue that followers’ OCB would have different target, either the individual or team/organization. There should be an alignment (as the literature says) between the identification and OCB target:

For instance, identification with leader often leads to OCB targeted at the leader as well. Or, as you said, when followers see leader as the extension of the organization, they might extend their OCB to organization as well. Identification with group likely leads to group-targeted OCB. And your operationalization (scales) should reflect such alignment. But I am not sure this is the case. For instance, the OCBI, is a mixed of both O and I-targeted scale.

Response to reviewer: We greatly appreciate your suggestion. We addressed your concerns about your points in the above section.  

Reviewer 2 Report

This manuscript introduces two intermediate variables: identification with the supervisor and social identification to examine the impact of authentic leadership on follower hope and organizational citizenship behavior. However, the following problems should be addressed before the paper will be published.

(1)  There are inconsistencies between line 24-25: “limited sample size and cultural specificity (South Korea), which curtails the scope of generalizability” and line 636-637: “the study's findings … can be applied to other countries” , but there is no further explanation.

(2)  The innovative description of the research work in the abstract is not detailed enough. In addition, it is not necessary to overstate in the abstract, such as line 24-28.

(3)  As the beginning of the manuscript, the introduction briefly introduces the background and purpose of the manuscript, the origin and the reality of the research requirements, as well as the general situation of existing research in related fields, explains the relationship between this research and previous work, research hotspots, existing problems and research significance, and leads to the theme of this paper to guide readers. Therefore, it is suggested to revise the introduction. A section should be listed to highlight the construction of theoretical models.

(4)  The manuscript has authenticated the mediation effect (H2, H3, H5, H6) by bootstrap analysis, but the robustness of H1 and H4 is still unknown.

(5)  In section 4.2, the management implications of this study should be explained, but the current formulation is not appropriate enough, for example line 631-635, and so on.

(6)  It is suggested that the title of section 3 be improved.

(7)  It is proposed that sections 4 and 5 be merged.

(8)  The manuscript does not determine “” which appears in line 438, 442, 445, 448 and 451.

(9)   Table 3 and 4 have the same table header “Regression analyses for mediation test”, which may make reader confused.

(10)  “B” in the fourth column of Table 3 should be replaced with “”.

(11) Please pay attention to the beautiful layout of the table and the specification of references.

Please check English: there are some minor mistakes.

Author Response

General Comment.

: We want to first thank you and the Reviewer for providing us with the opportunity to revise and resubmit our research. It is clear that you and the reviewers gave this manuscript (behavsci-2470103) very careful attention, and we have tried to reciprocate by addressing all of the various comments to the fullest extent possible. Below we have reproduced reviewers’ feedback in bold italics, followed by our response (blue color) in regular font. We carefully read the three reviewers’ comments and addressed and/or responded to each comment. We greatly appreciate this opportunity to revise and resubmit our manuscript. We welcome your continued feedback as we strive to make our manuscript as helpful to the field as possible.

Reviewer: 2

Recommendation:

This manuscript introduces two intermediate variables: identification with the supervisor and social identification to examine the impact of authentic leadership on follower hope and organizational citizenship behavior. However, the following problems should be addressed before the paper will be published.

Comments:

  1. There are inconsistencies between line 24-25: “limited sample size and cultural specificity (South Korea), which curtails the scope of generalizability” and line 636-637: “the study's findings … can be applied to other countries” , but there is no further explanation..

Response to reviewer: We appreciate your feedback. We deleted the sentence ‘limited sample size and cultural specificity (South Korea), which curtails the scope of generalizability in 24-25 line for the consistence. Many thanks.

  1. The innovative description of the research work in the abstract is not detailed enough. In addition, it is not necessary to overstate in the abstract, such as line 24-28.

Response to reviewer: Thank you for your time and efforts in reviewing our manuscript. We appreciate your feedback on the abstract, particularly your suggestion to provide more detailed information regarding our innovative research work.

We have revised the abstract, incorporating a more thorough description of our study and its contributions. The revised abstract better represents the work by detailing the theoretical foundations of the research, the data sources used, and the specific conclusions drawn from our study. It also provides a clearer explanation of the mediating roles of identification with the supervisor and social identification in the relationship between perceived authentic leadership and follower outcomes, namely hope and OCB.

We have endeavored to strike a balance between highlighting the significant contributions of our study and maintaining an appropriate level of modesty. The revised abstract avoids overstating our findings, while still emphasizing the important theoretical and practical implications of the research. Additionally, it acknowledges the limitations of our study and suggests future directions for research.

We believe these revisions have improved the quality of the abstract and better reflect the overall contributions of the manuscript. Once again, we appreciate your valuable feedback.

  1. As the beginning of the manuscript, the introduction briefly introduces the background and purpose of the manuscript, the origin and the reality of the research requirements, as well as the general situation of existing research in related fields, explains the relationship between this research and previous work, research hotspots, existing problems and research significance, and leads to the theme of this paper to guide readers. Therefore, it is suggested to revise the introduction. A section should be listed to highlight the construction of theoretical models.

Response to reviewer: We appreciate your insightful comments and valuable suggestions regarding our manuscript. We agree with your recommendation to better articulate the background, purpose, and significance of the research, as well as the relationship between this work and previous research. We also acknowledge the importance of clearly outlining the construction of our theoretical models.

We have revised the introduction accordingly, emphasizing the two main streams of authentic leadership research and elaborating on the focus of our research within this context. We then specify our research question related to the mechanisms through which authentic leadership influences follower outcomes, emphasizing the understudied role of follower identification. We further discuss the concept of follower identification, presenting it in the broader context of the authentic leadership–follower outcome relationship.

We also detailed the variables we are investigating: identification with the supervisor and social identification. In this revised version, we better explain how these constructs fit into the existing body of knowledge, and we provide clear theoretical arguments for our propositions. We have also clarified the unique contribution of our study, highlighting our examination of the role of hope and OCB as outcomes influenced by authentic leadership and the investigation of these relationships within a South Korean context.

Our changes in response to your comment have considerably improved the clarity of the manuscript and the rationale for our study. We believe that these changes have helped us better situate our research within the broader scholarly conversation on authentic leadership and underscore the significance of our contributions. Thank you for your time and for helping us improve our manuscript.

  1. The manuscript has authenticated the mediation effect (H2, H3, H5, H6) by bootstrap analysis, but the robustness of H1 and H4 is still unknown.

Response to reviewer: Thanks for your comment and feedback. We added some statistical numbers about direct effect of authentic leadership on follower identifications in the Table 5 using PROCESS Macro in SPSS.

  1. In section 4.2, the management implications of this study should be explained, but the current formulation is not appropriate enough, for example line 631-635, and so on.

Response to reviewer: Thank you for your valuable comments and suggestions regarding our manuscript. We appreciate your time and effort in reviewing our work and providing insights that have undoubtedly helped us improve the quality of our paper.

Regarding your comment on the management implications of our study, we agree that our previous formulation may not have adequately conveyed the practical applications of our findings. As such, we have undertaken significant revisions to provide a clearer, more precise explanation of the practical implications of our research.

To address this, we have expanded on the importance of innovative leadership development tools and the role of authentic leadership in fostering follower identification. We have also emphasized the significance of open dialogues between leaders and their teams, as these interactions often lead to higher levels of identification.

Furthermore, we have reiterated the role of hope in the workplace, corroborating prior research on its positive influence on employee performance. In the revised abstract, we highlight how authentic leaders can foster hope and resilience among employees, a point especially relevant in the context of post-pandemic recovery.

The revised abstract also offers more depth on the mediating roles of identification with the supervisor and social identification in the relationship between authentic leadership and follower outcomes, thereby providing a more comprehensive understanding of the dynamics of leadership and its impacts on followers.

Lastly, we have revised the abstract to emphasize that while our study primarily focuses on South Korean business organizations, the lessons learned about authentic leadership and follower identification can apply to countries with diverse cultural norms, such as Japan, China, or Singapore.

We believe these revisions address your concerns and appreciate your guidance in enhancing the clarity and comprehensiveness of our manuscript. We look forward to any further suggestions you may have.

  1. It is suggested that the title of section 3 be improved.

Response to reviewer: Thanks for your feedback. To adequately include the whole content, we have revised the title of “method and statistical analysis”. Many thanks.

  1. It is proposed that sections 4 and 5 be merged.

Response to reviewer: Thanks for your comment. As per requested, we have merged the section 4 and 5. Many thanks.

  1. The manuscript does not determine “” which appears in line 438, 442, 445, 448 and 451.

Response to reviewer: Thanks for your comment and feedback. We delete “ “ to avoid the potential confusion for the readers.

  1. Table 3 and 4 have the same table header “Regression analyses for mediation test”, which may make reader confused.

Response to reviewer: Thanks for your comment. We have revised the title of table 3 and 4 including ‘dependent variable of identification with the supervisor and social identification respectively’. Many thanks.

  1. “B” in the fourth column of Table 3 should be replaced with “.

Response to reviewer: Thanks for your comment and feedback. We have replaced with “β”.

  1. Please pay attention to the beautiful layout of the table and the specification of references.

Response to reviewer: Thanks for your comment and feedback. We have revised the table layout and ensured that the in-text citations and references adhere to the manuscript guidelines.

Reviewer 3 Report

This paper provides a comprehensive exploration of the relationship between authentic leadership and employee outcomes, specifically focusing on hope and organizational citizenship behavior. The authors have conducted a broad literature review, examining numerous studies to construct their hypotheses.

In their discussion, the authors propose that leadership development programs should prioritize cultivating authentic leaders. These leaders, they argue, can inspire their followers to engage in positive behaviors by enhancing their identification with the supervisor and social identification. This argument supports Hypotheses 1 and 4. Furthermore, the study underscores the role of hope in leadership, highlighting its positive correlation with follower motivation and goal attainment, thereby supporting Hypothesis 2.

The paper also offers practical implications for leadership development and fostering resilience in the workplace, adding to its value.

However, I have two recommendations for the authors to consider:

Firstly, the study was conducted in Korea, which limits its applicability to different cultural contexts. The authors suggest on page 4, line 148, that their findings can be generalized across cultures. I recommend removing this claim as it promises a conclusion that the study is not equipped to provide.

Secondly, the abstract could benefit from a rewrite. An effective abstract should include: (1) a statement of the research purpose, (2) a brief description of the methodology, (3) a summary of the main findings, and (4) a discussion of the conclusions drawn from the research. While limitations and implications are important, they are typically discussed at the end of the paper and may not be necessary in the abstract. Including them might deter readers from reading the full paper.

Author Response

General Comment.

: We want to first thank you and the Reviewer for providing us with the opportunity to revise and resubmit our research. It is clear that you and the reviewers gave this manuscript (behavsci-2470103) very careful attention, and we have tried to reciprocate by addressing all of the various comments to the fullest extent possible. Below we have reproduced reviewers’ feedback in bold italics, followed by our response (blue color) in regular font. We carefully read the three reviewers’ comments and addressed and/or responded to each comment. We greatly appreciate this opportunity to revise and resubmit our manuscript. We welcome your continued feedback as we strive to make our manuscript as helpful to the field as possible.

Reviewer: 3

Recommendation:

This paper provides a comprehensive exploration of the relationship between authentic leadership and employee outcomes, specifically focusing on hope and organizational citizenship behavior. The authors have conducted a broad literature review, examining numerous studies to construct their hypotheses.

In their discussion, the authors propose that leadership development programs should prioritize cultivating authentic leaders. These leaders, they argue, can inspire their followers to engage in positive behaviors by enhancing their identification with the supervisor and social identification. This argument supports Hypotheses 1 and 4. Furthermore, the study underscores the role of hope in leadership, highlighting its positive correlation with follower motivation and goal attainment, thereby supporting Hypothesis 2.

The paper also offers practical implications for leadership development and fostering resilience in the workplace, adding to its value.

Comments:

  1. Firstly, the study was conducted in Korea, which limits its applicability to different cultural contexts. The authors suggest on page 4, line 148, that their findings can be generalized across cultures. I recommend removing this claim as it promises a conclusion that the study is not equipped to provide.

Response to reviewer: We greatly appreciate your suggestion. As per requested, we removed that sentence and revised the abstract to address your feedback. Many thanks.

  1. Secondly, the abstract could benefit from a rewrite. An effective abstract should include: (1) a statement of the research purpose, (2) a brief description of the methodology, (3) a summary of the main findings, and (4) a discussion of the conclusions drawn from the research. While limitations and implications are important, they are typically discussed at the end of the paper and may not be necessary in the abstract. Including them might deter readers from reading the full paper.

Response to reviewer: point about refining the abstract to better include the research purpose, methodology, main findings, and conclusions is well-taken and appreciated. In response to your feedback, we have restructured the abstract to succinctly present the research's purpose, offer a brief overview of the methodology used, highlight the primary findings, and discuss the conclusions drawn from the research, without extending into the discussion of limitations and implications.

Below is the revised abstract:

"Authentic leadership's influence on follower outcomes through the mediating roles of identification with the supervisor and social identification forms the core purpose of this research. By studying this less-explored relationship within leadership studies, we aim to elucidate how these factors interrelate within the context of follower hope and organizational citizenship behavior (OCB). Using a quantitative methodology, we gathered and analyzed data from a sample of 241 employees across various South Korean businesses. Our main findings reveal that a follower's identification with the supervisor significantly mediates the relationship between perceived authentic leadership and follower outcomes such as hope and OCB. Concurrently, the study found that strengthening employee identification with their work group positively enhances these outcomes. From these findings, we conclude that authentic leadership can effectively drive follower identification, fostering beneficial outcomes such as hope and OCB. It also suggests that workplaces that promote authentic leadership and a culture of strong supervisor and social identification can effectively enhance overall employee well-being and organizational performance."

We believe that this revised abstract now effectively captures the essence of the research, highlights our methodology and primary findings, and presents the conclusions more succinctly. We hope that this modification meets your expectations and improves the quality of our paper. Thank you once again for your invaluable feedback. We look forward to any additional comments or suggestions you may have to further improve our manuscript.

Round 2

Reviewer 1 Report

I do appreciate authors' effort in revising the manuscript. The conceptual development and organization are improved to my satisfaction.

Moderate language editing needed.